# It Runs in the Bromodomain Family: Speckled Proteins (SP) Play a Role in the Antitumor Immune Response in Solid Tumors

**DOI:** 10.3390/ijms24010549

**Published:** 2022-12-29

**Authors:** Monika Anna Rosochowicz, Julia Maria Lipowicz, Marianna Iga Karwacka, Julia Ostapowicz, Malgorzata Cisek, Andrzej Adam Mackiewicz, Patrycja Czerwinska

**Affiliations:** 1Undergraduate Research Group “Biobase”, Poznan University of Medical Sciences, 61-701 Poznan, Poland; 2Radiobiology Laboratory, Greater Poland Cancer Centre, 61-866 Poznan, Poland; 3Department of Orthopedics and Traumatology, Poznan University of Medical Sciences, 61-545 Poznan, Poland; 4Department of Histology and Embriology, Poznan University of Medical Sciences, 60-781 Poznan, Poland; 5Department of Cancer Immunology, Poznan University of Medical Sciences, 61-866 Poznan, Poland; 6Department of Electroradiology, Poznan University of Medical Sciences, 61-866 Poznan, Poland; 7Department of Diagnostics and Cancer Immunology, Greater Poland Cancer Centre, 61-866 Poznan, Poland

**Keywords:** BrD, bromodomain protein, TCGA, immune infiltration, speckled protein

## Abstract

Cells and immune cells in the extracellular matrix: Depending on the tumor type and variety of TAAs (tumor-associated antigens), immune infiltrates are composed of many different subpopulations of immune cells. Epigenetic changes are also considered to be characteristic of cancer. Epigenetic factors taking part in the regulation of gene expression include the VII group of bromodomain proteins (BrD)—SP-family proteins. Here, we used transcriptomic data from the TCGA database, as well as immunological evidence from ESTIMATE, TIP, and TIMER2.0 databases for various solid tumor types and harnessed several publicly available bioinformatic tools (such as GSEA and GSCA) to demonstrate mechanisms and interactions between BrD proteins and immune infiltrates in cancer. We present a consistently positive correlation between the SP-family genes and immune score regardless of the tumor type. The SP-family proteins correlate positively with T cells’ trafficking and infiltration into tumor. Our results also show an association between the high expression of SP family genes and enriched transcriptome profiles of inflammatory response and TNF-α signaling via NF-κβ. We also show that the SP-family proteins could be considered good predictors of high immune infiltration phenotypes.

## 1. Introduction

Healthy somatic cells present a homeostatic balance of chemical reactions and signaling pathways. Any disruption in those mechanisms may trigger cells’ pathological transformation into a cancer cell. Genetic changes upregulate proliferation and enhance resistance to apoptosis-inducing signals, promoting tumor growth [1,2]. In addition to genetic dysregulation, tumor cells exhibit an altered epigenome through changes in DNA methylation, chromatin structure, and histone modifications [3]. These changes also include proteins that initiate or read histone post-translational modifications (PTMs), such as methylation or acetylation. Abnormal distribution of histone modifications results in relaxation or compaction of different regions of chromatin, thus activating/silencing different genes [4].

One of the epigenetic factors taking part in the regulation of gene expression is a group of Bromodomain proteins (BrD) that are considered to be the first discovered chromatin readers [5]. BrD are readers of acetylated lysine (Kac) on nucleosomal histones: they recognize modified histone and recruit effector proteins capable of chromatin remodeling or transcription activation [5,6]. Bromodomain itself is a structural motif consisting of 110 AA in the form of four alpha helices (αZ, αA, αB, and αC). Kac modifications are recognized by interhelical loops αZ-αA (ZA loop) and αB-αC (BC loop), which create a hydrophobic pocket. BC loop is highly conserved, although the rest of the sequence varies among BrD proteins [5,6,7].

There are 41 BrD proteins which have been categorized into groups based on structural similarities. They present a diverse array of activities, chromatin remodeling, and transcription initiation or elongation, among others. Because of this, Zaware and Zhou [5] proposed a function-based classification consisting of nine subgroups (Table 1).

Group I includes histone acetyltransferases, group II includes histone methyltransferases, group III includes chromatin remodeling factors, group IV includes AAA ATPase proteins, group V includes BET family transcriptional coactivators, group VI includes E3 SUMO/ubiquitin ligases, group VII includes SP-family proteins of PML nuclear bodies, group VIII includes transcriptional corepressors, and group IX includes WD-repeat proteins [5,8]. Structural and functional classifications show partial overlap, but functional classification more clearly presents BrD proteins’ diversity and quantity.

In the course of our analysis, group VII (SP-family proteins of PML nuclear bodies) was distinguished from the other BrD proteins. The Speckled Protein (SP) family consists of four proteins: SP100, SP110, SP140, and SP140-like protein (SP140L) [9]. They are associated with promyelocytic leukemia nuclear bodies (PML-NBs) and multiprotein complexes present in a variety of diseases, including primary biliary cirrhosis and acute promyelocytic leukemia [9]. SP family members share a similar structure, and interactions with DNA are carried out by a SP100, Aire, NucP41/P75 and Deaf (SAND) domain, a plant homeodomain (PHD), and bromodomain (BRD). The SAND domain is associated with chromatin-dependent transcriptional regulation through direct DNA binding [10]. PTMs’ reader functions of SP-family proteins are carried out mostly by PHD and BRD. PHD recognizes unmethylated lysine in H3 (H3K4me0) and BRD stabilizes PHD fold and reads histone acetylation, creating a dual-reader module [11,12].

Interestingly, a similar structure to the SP-family proteins is shown by AIRE protein [5]. AIRE is an autoimmune regulator with the ability to bind and regulate gene transcription. It comprises the same domains as SP proteins, with the exception of BRD, which is substituted by a second PHD [13]. Due to structural similarities and simultaneous lack of BRD, we analyzed AIRE and compared it with the results obtained from the SP analysis.

As AIRE was assigned to be a structural comparison to the SP-family proteins, we chose another BrD protein to be a functional comparison. ATAD2 is a member of group IV (AAA ATPase proteins) and a histone acetylation reader. It recognizes di-acetylation of lysins in H4 (H4K5acK12ac), which is characterized as an indicator of the DNA replication site [14]. ATAD2 has two domains with ATPase activity and BRD at the C-terminus. ATPase domains are responsible for the multimerization of the complex, while BRD acts as a PTM reader [6,14].

The dysregulation of BrD proteins has been repeatedly acknowledged in different types of cancer. TRIM28 overexpression in breast cancer was correlated with lymph node metastasis and shorter overall survival (OS) [15]. BRD4 was reported to be involved in the development of leukemia, prostate cancer, and breast cancer, as confirmed by BET-inhibitor (BETi) studies [16]. It shows that BrD proteins in cancer should be further investigated.

Epigenetic changes are considered to be enabling characteristics of cancer. One of the emerging hallmarks of cancer is the capacity to avoid immune destruction [2]. The tumor microenvironment (TME) consists of tumor, stromal, and immune cells in the extracellular matrix [17]. Properly functioning immune cells show the ability to recognize tumor cells because tumor cells express ligands (TAAs, tumor-associated antigens) on the cell surface that are not present in the surrounding tissue [17]. TAAs are then presented by antigen-presenting cells (APCs), resulting in the initiation of T cells’ expansion with matching T-cell receptors (TCRs). Depending on the tumor type and variety of TAAs, immune infiltrates are composed of many different subpopulations of immune cells [17]. Tumor-infiltrating lymphocytes (TILs) determine tumor growth, response to treatment, and other clinical outcomes. Prolonged OS was defined in advanced-stage ovarian cancer patients with TILs present [18]. Similarly, in patients with colorectal cancer, TILs were characterized as an independent prognostic factor of OS and recurrence-free survival (RFS) [19]. In contrast to these findings, patients with renal carcinoma with TILs present shorter RFS compared to patients without them [20].

As mentioned before, epigenetic changes and immune infiltrates are considered characteristics and hallmarks of cancer [2]. Little is known about the mechanisms and interactions between the BrD proteins and immune infiltrates in cancer. Here, we examine publicly available data from The Cancer Genome Atlas (TCGA) to determine the association between chosen BrD protein families and immune infiltration defined by different databases.

We used Estimation of Stromal and Immune cells in Malignant Tumor tissues using Expression (ESTIMATE) data to determine whether any of the BrD protein families presented a unified correlation between expression and defined score (either stromal, immune, or estimate score). Based on the outcome of this analysis, we concentrated on SP-family proteins of PML nuclear bodies. Using The Gene Set Cancer Analysis (GSCA), we analyzed if the expression of SP-family genes correlates with the abundance of selected populations of immune cells. Then we compared the immune cell infiltration results with those obtained by using the seven-step cancer-immunity cycle from the Tracking Tumor Immunophenotype (TIP) database. The Gene Set Enrichment Analysis (GSEA) was carried out to demonstrate alterations in transcription profiles associated with SP-family genes across 21 tumor types. Publicly available data from the TISIDB were used to determine whether SP-family genes’ expression affects immunomodulators across 27 tumor types. Finally, the TIMER2.0 database was used to analyze the effect of immune infiltration on patients’ survival prognosis, while taking into account SP-family genes’ expression across 27 tumor types.

## 2. Results

### 2.1. Correlation of the Expression of SP Family Members and Scores from the ESTIMATE Database Is Consistently Positive across All Tumor Types

Here, we analyzed the correlation between the BrD family members’ expression and three estimate scores across 21 TCGA tumor types. The stromal score is the indicator of stroma’s presence in tumor tissue, the immune score represents the infiltration of immune cells in tumor tissue, and the estimate score indicates tumor purity [21]. As shown in Figure 1A, we found a striking positive correlation between the SP subfamily of Bromodomain proteins, namely SP100, SP100, SP140, and SP140L, and the estimate score across all analyzed tumor types. Of the entire SP family, SP140L expression was the least correlated with the estimate score. However, except for a few tumors with no statistical significance or weak negative correlation, the trends for this gene agreed with the rest of the SP family. We also observed a strong positive correlation between SP-family genes’ expression and other scores provided by the ESTIMATE database—immune and stromal—regardless of the tumor type (Figure 1B,C and Appendix A). As for other members of the Bromodomain family, we did not observe any consistent association with tested scores across analyzed TCGA tumor types.

### 2.2. Gene Set Cancer Analysis Shows a Strong Correlation between SP Family Expression and Immune Cell Infiltration

Because the abundance and distribution of immune cells, such as T-cell subsets, play a crucial role in cancer therapy, we performed the GSCA by using the Immune Cell Abundance Identifier (ImmuCellAI) algorithm for SP family members across 28 TCGA solid tumor types. ImmuCellAI is performed to precisely estimate the abundance of 24 immune cell types from gene-expression data [22]. Firstly, we focused on the skin cutaneous melanoma (SKCM), an immunogenic tumor whose relationship with immune cells resident in the microenvironment significantly influences cancer cell proliferation, progression, and metastasis. We focused on this type of cancer because of the greatest completeness of the collected data. To confirm the result obtained, the same analyses were performed for head and neck squamous cell carcinoma (HNSC) and bladder urothelial carcinoma (BLCA), and figures showing the results obtained are included in Appendix A Appendix A. As shown in Figure 2A, we observed a high negative correlation between SP family members’ expression and neutrophil abundance in SKCM. On the other hand, for B cells and distinct T-cell subpopulations, we observed mostly positive correlations in SKCM, with the *SP140* gene exhibiting the strongest associations. Furthermore, across other solid tumor types, only *SP140* showed a strong positive correlation with distinct T-cell subpopulations, followed by negative associations with a neutrophil population (Figure 2B). *SP100* was mostly negatively correlated with B-cell abundance, while the rest showed a positive association. The correlation of SP genes and other immune cell cohorts, namely T follicular helper cells (Tfh), Natural Killer (NK) cells, macrophages, induced regulatory T cells (Tregs), CD8+ T cells, and CD4+ T cells, was mostly positive (Figure 2C and Appendix A). Moreover, we looked at the association between the abundance of immune cell populations and *AIRE* expression-the gene which shares high homology with SP family members, although it doesn’t possess the Bromo domain [13]. Correspondingly with SP family members, *AIRE* was negatively correlated with neutrophil abundance. However, it also exhibited a negative correlation with the macrophage scores. To highlight that the abovementioned associations with the immune-cell infiltrations are specific to the SP sub-family of Bromodomain proteins, we selected the *ATAD2* gene as a representative of other BrD proteins and demonstrated that the association between *ATAD2* gene expression and immune cell infiltration in solid tumors was not so consistent and significantly varied between tumor types and immune-cell types.

Therefore, the observed positive associations between distinct T-cell subtypes and the expression of SP family members are specific to the SP subfamily of BrD proteins.

Next, we validated the observed results with an additional database, the TIP database, which uses a single-sample GSEA (ssGSEA) and CIBERSORT methods for analyzing the status of antitumor immunity. Indeed, we found that the expression of SP-family genes and *AIRE* gene was correlated mostly positively with the presence of specific immune-cell subtypes, while the *ATAD2* showed opposite trends (Figure 2D–F and Appendix A).

### 2.3. Significant Enrichment of Immune-Related Hallmark Gene Signatures in SP-Family-Associated Transcriptome Profiles

The SP family is known to be associated with various autoimmune, inflammatory, or infectious diseases in humans [9]. However, the precise roles of each protein are still unknown. To better understand the meaning of these proteins in terms of the immune response, we performed the GSEA and compared the SP family-associated transcriptome profiles with priori-defined gene signatures from the MSigDB. As shown in Figure 3A, we found that, in various cancers (represented here by SKCM, BLCA, and HNSC), the transcription profiles associated with SP-family members were significantly enriched with the Inflammatory Response Hallmark term. The range of normalized enrichment score (NES) for this signature varied depending on the gene and type of cancer, with the highest values observed for *SP110* in HNSC. As for *AIRE* or ATAD2-associated transcriptome profiles, we did not obtain statistically significant results.

We also investigated the enrichment of other immune-related gene signatures (Figure 3B,C and Appendix A) across 27 tumor types. A vast majority of them were significantly enriched in SP-family-associated transcriptome profiles, with the exception of the TGF-β signature, which was depleted in one (*SP110*) and two (*SP140*) tumor types (however, the results for this signature were mainly statistically not significant).

### 2.4. Better Survival Prognosis Is Linked with Immune Cells’ Infiltration Associated with SP-Family Expression

Furthermore, we investigated the association between survival prognosis and immune cells’ infiltration and the associated with SP-family genes’ expression. We chose T cells CD4+ and CD8+ and also macrophages because these three cell types can represent TIICs and generally stand for infiltration in the immune response. As shown in Appendix A, the survival prognosis in SKCM depends on the abundance and type of TIICs. An analysis of the SP-family genes’ expression may help to predict the profusion of tumor infiltration (Figure 4). The AUC values denote accuracy for that prediction; out of all analyzed genes, the highest AUC values were observed for SP140 (for CD4+ T-cell subpopulation, AUC = 0.754; for CD8+ T-cell subpopulation, AUC = 0.836; for macrophage subpopulation, AUC = 0.851) and the lowest for the control gene, *AIRE* (there were no significant results for *ATAD2*), suggesting that the SP subfamily of Bromodomain proteins possesses some prognostic value.

### 2.5. SP-Family Genes’ Expression Is Positively Correlated with Immunoinhibitors’ Abundance across 27 Tumor Types

Next, we examined the association between SP-family genes and selected immunomodulators, namely *PCDC1* (encoding for PD1), CTLA4, and *CD274* (encoding for PD1-L1). We mostly observed significant positive correlations between PDCD1 and CTLA4 vs. each of the SP family members, with *SP140* exhibiting the most robust associations (Figure 5). As for CD274, only *SP140L* correlated negatively in five cancer types. The results for *AIRE* were mostly insignificant and varied depending on the immunomodulator and cancer type. *ATAD2* expression was correlated mostly negatively with PDCD1 and CTLA4 abundance and, interestingly, mostly positively correlated with CD274 abundance.

All of the results described above suggest that Speckled Protein family members are strongly associated with the immune cells’ abundance and infiltration, regardless of the tumor type. Their high expression is also correlated with better survival prognosis and the aforementioned immunomodulators’ abundance.

## 3. Discussion

This is the first comprehensive report of the association between all known BrD proteins and immune-cell-infiltration status across different types of solid tumors. So far, only several BrD proteins have been analyzed for their role in the antitumor immune response. Specifically, BRD1/2/3/4/7/8/9 proteins were analyzed in breast [23,24], liver [25], lung [26], ovarian [27], prostate [28], pancreatic [29], and renal cancer [30]. Here, we used transcriptomic data from the TCGA database [31], as well as immunological evidence from the ESTIMATE [21], the TIP [32], and the TIMER 2.0 databases [33], for various solid tumor types and harnessed several publicly available bioinformatic tools (such as the GSEA and the GSCA) to demonstrate the following:BrD family members show a differential correlation between gene expression and immune-cell-infiltration status;SP-family genes (*SP100*, *SP110*, *SP140*, and *SP140L*) exhibit a consistently positive correlations with distinct immune scores, regardless of the tumor type;Based on the TIP database and GSCA data, the SP family shows a mostly positive correlation with T cells’ trafficking and infiltration into tumors. A high expression of SP-family genes is observed especially in solid tumors with great T-cell and macrophage abundance, and to a certain extent in tumors with enriched B-cell or neutrophil infiltrates;The transcriptome profiles associated with high expression of the SP-family genes are significantly enriched with allograft rejection, inflammatory response, IL2-STAT5, IL6-JAK-STAT3, TNF-α signaling via NF-κβ, and interferon-signaling signatures;SP genes’ expression might serve as a marker of immune cells’ invasion, while indirectly informing the chance of infiltration-dependent survival in various solid tumors.SP-family genes, especially *SP140*, show a highly positive correlation with the abundance of PDCD1 (PD-1), CTLA4, and CD274 (PD-L1) immunoinhibitors;As bromodomain present in SPs’ proteins is easily targetable, it may be an effective target for small molecule therapeutics to enhance immunotherapy outcomes.

The TME is a highly heterogeneous, complex ecosystem composed of malignant, stromal, and immune cells. The composition of TIICs influences tumor progression and can serve as a prognostic factor for anticipating the response to treatment and survival in cancer patients [34,35]. This is the first report that comprehensively analyzes the association of BrD proteins with the presence of TIICs across numerous types of solid tumors based on previously reported statements on the significance of TIICs in tumor progression [35].

To date, only several members of the BrD family were directly connected with immune cell infiltration of solid tumors. Bromodomain-containing protein 4 (BRD4), a BET family, is especially known to influence the tumor immune surveillance. *BRD4* gene expression was shown to have an immunomodulatory function in breast cancer and was positively correlated with the levels of infiltrating B cells, CD8 + T cells, CD4+ T cells, macrophages, neutrophils, and dendritic cells (DCs) [36]. Bromodomain-containing protein 1 (BRD1) is a negative regulator of T and NK cell activity in high-grade serous ovarian cancer (HGSC) [27]. In hepatocellular carcinoma, the expression of some BrD-family coding genes (*BRD1/2/3/4/7/8/9*) were significantly positively correlated with the immune-infiltrating degree of B cells, CD8+ T cells, CD4+ T cells, macrophages, neutrophils, and DCs [25].

We have recently reported that the significant upregulation of the BrD family members *ATAD2* and *SMARCA4* and downregulation of *SMARCA2* are associated with enriched cancer stemness [37]. A higher cancer-stem-cell-like phenotype is known to be negatively correlated with immune-cell infiltration in the tumor microenvironment [38]. In line with these reports, in the current study, *SMARCA4* remained negatively correlated with cancer immune cell infiltration. However, *ATAD2* and *SMARCA2* were differentially associated with immune surveillance in various solid tumors.

The SP-family genes, which were highlighted here as prognostic factors associated with increased TIICs levels, are central chromatin regulators involved in immune-cell identity and function control [9]. *SP100* and *SP110* genes are expressed in non-immune cells, as well as in innate and adaptive immune cell lineages, whereas *SP140* is exclusively expressed in immune cells, with greatest abundance in mature B cells, DCs, macrophages, and granulocytes and low amounts in hematopoietic stem cells (HSCs), monocytes, and T cells [9,39].

SP100 protein isoforms comprise PML-NBs, nuclear substructures which regulate genome stability, alternative lengthening of telomeres, epigenetic regulation of chromatin, antiproliferation, senescence, apoptosis, and antiviral immunity [40]. SP140 is a key regulator of transcriptional programs that support macrophage activation in response to cytokines and microbes. SP140 is also responsible for the repression of HOX genes, which are known to promote the stem-like state in HSCs and hamper macrophage differentiation [39]. Depletion of *SP140* in mouse and human macrophages resulted in the significant downregulation of a number of inflammatory or innate immune genes causing ineffective innate immunity [39]. However, until now, the role of SPs in the control of TIICs levels remained largely unknown.

Here, we demonstrate a significant correlation between *SP100*, *SP110*, *SP140*, and *SP140L* genes’ expression and immune cell populations in the tumor microenvironment. SP family members’ expression was positively correlated with neutrophil, B-cell, Th2-cell, NK-cell, macrophage, Treg, CD8+ T-cell, and CD4+ T-cell infiltration based on the TIP database and the GSEA analysis. This relationship may have some predictive value, as the cellular composition of the tumor mass influences tumor growth and patient survival. Indeed, the association between selected immune cell types and the hazard ratio of death showed that enriched CD8+ T cells’ infiltration is favorable in LGG and SKCM; however, it is associated with poor prognosis in ACC, GBM, and UVM. A high level of CD4+ T cells’ infiltration is favorable in HNSC, PAAD, and SARC, but it is associated with poor prognosis in COAD, KICH, and LGG. A high abundance of macrophages in the tumor microenvironment is associated with a poor prognosis in BLCA, BRCA, KICH, LGG, LIHC, MESO, and STAD.

The SP-family genes have not only a great potential as biomarkers, but also their bromodomains act as easily targetable modules for small molecule inhibitors. Since the development of the first small molecule BETi [7], a variety of efficient molecules that bind selectively to bromodomains have been developed [41]. As all bromodomain domains share a conserved fold comprising a left-handed bundle of four alpha helices (αZ, αA, αB, and αC), connected by diverse loop regions that contribute to substrate selectivity [7], it is potentially achievable to design such molecules that will selectively bind to any of the BrD.

An effective response to immunotherapy requires proper work of the integrated immune system in cooperation with the drug used. Damage in any step of the immune response is one of the most disastrous causes of immunotherapy failure [42]. Targeting epigenetic modulators, such as SP-family proteins, with small molecule inhibitors could restore normal immune response and immune cell infiltration into the tumor. Moreover, since SP proteins emerged as modulators of immunoinhibitors PD-1 and PD-L1, targeting SP proteins may additionally bolster the immune checkpoint blockade. In the future, combining the SP proteins’ inhibitors with immunotherapy may be a new approach for patients showing high resistance to immunotherapy or in those neoplasms where immunotherapy shows unsatisfactory results. This theory is additionally supported by the fact that the expression of SP genes is associated with many phenomena of great importance for the cancer immune response. We present a significant enrichment of allograft rejection, inflammatory response, IL2-STAT5 signaling, IL6-JAK-STAT3 signaling, tumor necrosis factor α (TNF-α) signaling via NF-κβ, and interferon signaling signatures in transcriptome profiles associated with high expression of SP-family genes.

One of the featured gene-set signatures was allograft rejection. Chemokines associated with allograft rejection are CXCR3 and CCR5 ligands, CXCL9, CXCL10, CCL4, and CCL5 [43]. These factors, among other proteins frequently upregulated during allograft rejection, are consistent with the predominance of Th1-cell polarization [43]. Th1 cytokines such as interferon γ (IFNγ), TNFα, or granulocyte/monocyte-colony-stimulating factor (GM-CSF) are implicated in tumor immunosurveillance and regulate cytotoxic activity [43]. Enriched allograft rejection, as well as INF-signaling and TNFα-signaling signatures, was found in solid tumors with a high expression of SP-family genes. Thus, it is possible that the upregulation of SPs correlates with a stronger antitumor response. However, Th1 polarization coupled with IFN-γ-mediated signaling may also reinforce the immunosuppressive character of tumor-associated macrophages (TAMs) [44].

The allograft-rejection-gene-set signature also includes class I human leukocyte antigens (HLA-I). The process of tumor antigens displayed by HLA-I determines the positive removal of cancer cells by the immune system [45]. The positive correlation of the SP genes with the HLA-I-group genes suggests that SPs may play a role in the immune response against cancer by enhancing cancer antigens’ presentation and trafficking of active CD8 + T cells into the tumor milieu.

The SP genes’ expression profile correlates with enriched IL2-STAT5 signaling. The STAT5 pathway is critical for activating lineage-specific enhancers and suppressing enhancers associated with alternative cell fates during CD4+ T-cell differentiation. IL-2 via STAT5 promotes Th1, Th2, Th9, and Treg differentiation and inhibits Th17 or Tfh cells differentiation [46]. From one point of view, the IL2-STAT5 activates CD4+ T-cells differentiation into cell populations favoring an antitumor response, such as Th1 and Th9 cells [46]. From the other point of view, the IL2-STAT5 pathway may elevate the immunosuppressive activity of Treg cells and compromise the immune response [47]. Hampering Treg cell responses is beneficial in various tumor immunotherapy approaches, as it sustains inflammatory conditions within the tumor environment [48].

IL-6 and JAK/STAT3 signaling and related inflammatory response genes were found to be enriched in solid tumors presenting high levels of SP-family genes’ expression. The IL6-JAK-STAT3 pathway promotes the proliferation, survival, invasiveness, and metastasis of tumor cells; induces the production of pro-angiogenic factors; and has a significant influence on TIICs [49]. STAT3 activation contributes to a highly immunosuppressive tumor microenvironment due to increased secretion of IL-10 and TGFβ, negative regulation of neutrophils, NK cells, effector T cells, and DCs and positive regulation of Treg cells and myeloid-derived suppressor cells (MDSCs) [49].

The NF-κB signaling pathway is a key coordinator of innate immunity and inflammation. Its activation induces the production of proinflammatory cytokines (such as TNFα, IL-1, IL-6, and IL-8) and adhesion molecules, which enables leukocyte recruitment to the site of inflammation [50]. NF-kB is also involved in antigen presentation during T-cell proliferation and differentiation and has a critical role in B-cell maturation and survival [51]. NF-κB participates both in tumor immunosurveillance [50], as well as possesses some pro-tumorigenic functions [51]. Nf-κB activates TNFα signaling in CAMs (also named M1). However, chronic inflammation may change TAMs phenotypes towards alternatively activated macrophages (AAMs, M2), which release mainly anti-inflammatory cytokines and cause immune suppression within the tumor microenvironment [50].

Therefore, we suggest that the cancer immune cell infiltration and polarization, as well as tumor immunosurveillance activation through SPs regulatory functions, may act as a double-edged sword, causing both antitumor immune response and cancerogenic inflammatory reaction. On the one hand, enriched SP levels correlate with the activation of an antitumor response mediated by IFNγ, TNFα, M1 macrophages, and Th1 and Th9 CD4+ T cells. On the other hand, upregulated SP-family genes’ expression is associated with the immunosuppressive functions of IL-10 and TGFβ, M2 macrophages, MDSCs, and Tregs, as well as the negative regulation of neutrophils, NK cells, and DCs. Despite the fact that our results are based on abundant data, it should be noted that, for some tumors, the amount of information was limited, and this is why some analyses were conducted for a lower number of solid tumor types. Additionally, the infiltration analysis did not include tumor subtypes. The correlation between immune cell invasion and specific tumor type could be an interesting follow-up. Lastly, the results shown in this manuscript have not been confirmed experimentally yet. The available knowledge about SP-family proteins is still superficial. Thus, molecular studies are indispensable in order to determine the exact role of SPs in immune cell infiltration of solid tumors.

## 4. Materials and Methods

### 4.1. TCGA Solid Tumor Types Selected for the Study

In the presented study, we selected 21 solid TCGA tumor types which have both (a) more than 50 samples and (b) available survival data. Access to TCGA online data is unrestricted and does not require patients’ consent or other permissions. 

### 4.2. TCGA Genomic Data

Genomic data for 27 solid TCGA tumors were directly downloaded from the cBioportal (www.cbioportal.org, accessed on 5 January 2022) [31] database.

### 4.3. Transcriptomic and Clinical Data

The RNA-sequencing-based mRNA expression data were directly downloaded from the cBioportal. RNASeq V2 from TCGA was processed and normalized by using RSEM [52]. Specifically, the RNASeq V2 data in cBioPortal correspond to the rsem.genes.normalized_results file from TCGA. The Spearman’s correlation was used for detection of co-expressed genes, with *p*-value < 0.05 and FDR < 0.01 as cutoffs. Differentially expressed genes (DEGs) were cut off at *p*-value < 0.05 and FDR < 0.05. We used cBioPortal database to acquire clinical data (sex, age of diagnosis, OS, grade, stage, OS status, OS time, DFS status, DFS time, and sample type).

### 4.4. Estimate

Using the Estimate database [21], we downloaded stromal, immune, and ESTIMATE scores for each sample across 21 TCGA solid tumor types. The obtained data were further applied to perform Spearman’s correlation [53] between BrD family members’ expression and scores from estimate based on high/low expression partition of patients, using the median as the cutoff point. We compared stromal, immune, and estimate scores for tumors and genes separately.

### 4.5. Gene Set Cancer Analysis

The association between BrD family members’ expression and lymphocytes infiltration across 21 solid tumor types was analyzed with the GSCA (http://bioinfo.life.hust.edu.cn/GSCA/#/, accessed on 15 January 2022) [54] platform, using the “Immune infiltration & mRNA expression’’ analysis module.

### 4.6. Tracking Tumor Immunophenotype

The TIP database [32] (http://biocc.hrbmu.edu.cn/TIP/, accessed on 1 March 2022) was used to analyze and visualize the status of the anticancer immunity and the proportions of tumor-infiltrating immune cells (TIICs) across a seven-step Cancer-Immunity Cycle for the selected 21 TCGA solid tumor types. We based our analysis on the Cancer-Immunity Cycle Steps IV and V, which stand for trafficking of immune cells to tumors and infiltration of immune cells into tumors, respectively. Based on T-cell CD4+, T-cell CD8+, and macrophage status from Step IV, we investigate the prognostic significance of SP family members and control genes related to the infiltration of relevant immune cell populations. The area under the curve (AUC) was calculated for the Receiver Operating Characteristics (ROCs) [55] curve. 

### 4.7. Gene Set Enrichment Analysis

The GSEA (http://www.broad.mit.edu/gsea/index.html, accessed on 10 March 2022) [56] was used to detect the coordinated expression of a priori defined groups of genes within the tested samples. Gene sets are available from the Molecular Signatures Database (MSigDB, http://www.broad.mit.edu/gsea/.msigdb/msigdb_index.html, accessed on 10 March 2022) [57]. We used hallmark (H) and ontology (C5) gene set collections. BrD family members’ gene lists in ranked list file format (*.rnk) for 21 solid tumor types [TCGA] and BrD proteins were imported to GSEA. The GSEA was run according to the default parameters: each probe set was collapsed into a single gene vector (identified by its HUGO gene symbol), permutation number = 1000, and permutation type = “gene-sets”. The FDR < 0.01 was used to correct for multiple comparisons and gene set sizes.

### 4.8. TISIDB

The association between BrD family members’ expression and the immunomodulators was assessed by using the TISIDB portal (http://cis.hku.hk/TISIDB/index.php, accessed on 5 March 2022) [58] among 30 tumor types. The correlation was calculated by using Spearman’s rank correlation coefficient (r).

### 4.9. TIMER 2.0

The TIMER 2.0 (http://timer.cistrome.org/, accessed on 20 February 2022) [33] database was used to explore immune infiltrates across 27 cancer types by association between immune infiltrates and BrD family gene expression and subsequently with clinical patients’ outcome. The effect of immune infiltration level on patient clinical outcome was assessed by using the Cox proportional hazard model and all available deconvolution methods (TIMER, EPIC, and CIBERSORT).

### 4.10. Statistical Analysis

Statistical analyses were carried out with GraphPad Prism 9.3.0 software (GraphPad Software, Inc., La Jolla, CA, USA). Multiple comparisons were performed with the ANOVA test. The correlation between two variables was assessed with Spearman’s rank correlation coefficient (r).

## 5. Conclusions

The conducted research shows a differential correlation between BrD family gene expression and immune cell infiltration status. Among all BrD protein groups, the SP-family genes show a universally positive association with the immune score regardless of the tumor type. The high expression profile of *SP100*, *SP110*, *SP140*, and *SP140L* corresponds with T cells’ infiltration in solid tumors. Collectively, we propose that the SP family may be used as a potential target for immunotherapeutics to improve treatment outcome.

## Figures and Tables

**Figure 1 ijms-24-00549-f001:**
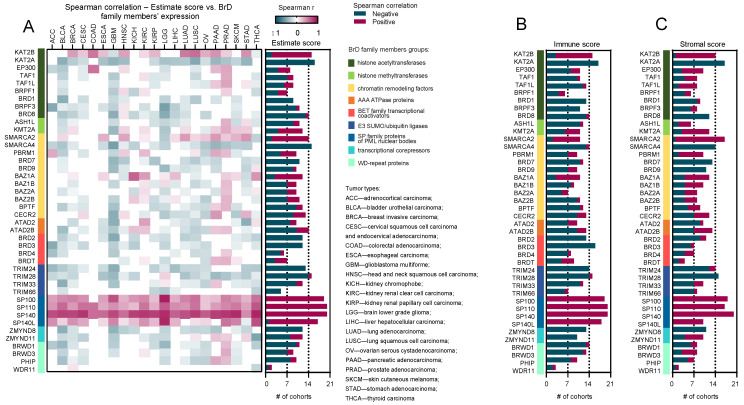
Estimate score signature associated with the expression of BrD family members. (**A**) The heatmap of Spearman’s correlation between the BrD family members expression and the estimate score among 21 solid tumor types in distinct TCGA cohorts with the number of TCGA cohorts (# of cohorts), showing positive (pink) or negative (dark blue) correlations between expression of BrD family genes and estimate score. Color on the heatmap denotes either expression is upregulated (pink) or downregulated (dark blue). White squares denote no statistical significance (*p* > 0.05). (**B**,**C**) Summarized number of TCGA cohorts with either positive (pink) or negative (dark blue) correlation between the BrD family members expression and immune (**B**) and stromal score (**C**).

**Figure 2 ijms-24-00549-f002:**
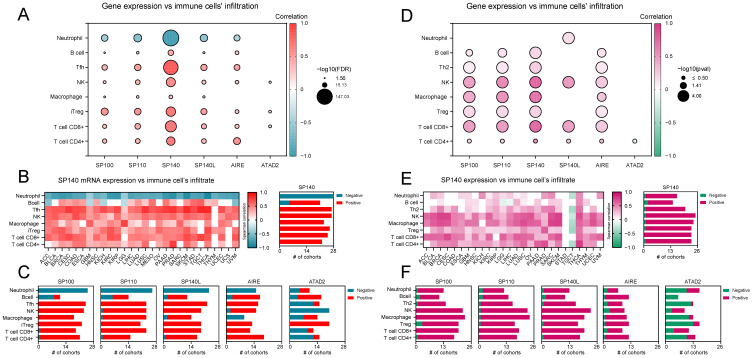
BrDs that belong to the SP subfamily are positively associated with tumor infiltration status. (**A**) The correlation between selected BrD family members’ expression and immune cells’ infiltration in SKCM, using the GSCA database (http://bioinfo.life.hust.edu.cn/GSCA/, accessed on 15 January 2022). Red and blue denote positive and negative correlations, respectively. Bubble size depicts statistical significance (−log10(FDR)). (**B**) The heatmap of all statistically significant correlations between SP140 expression and immune cells’ infiltration across 28 TCGA solid tumor types, using the GSCA database. Red and blue denote positive and negative correlations, respectively. The number of TCGA cohorts correlated positively (red) or negatively (blue) is presented on the right panel. (**C**) The number of TCGA cohorts showing positive (red) or negative (blue) correlations between the expression of selected markers and immune cell infiltration using the GSCA database. (**D**) The correlation between selected BrD family members’ expression and immune cells’ infiltration in SKCM, using the TIP database (http://biocc.hrbmu.edu.cn/TIP/pancancerAnalysis.jsp, accessed on 1 March 2022). Pink and green denote positive and negative correlations, respectively. Bubble size depicts statistical significance (−log10(pval)). (**E**) The heatmap of all statistically significant correlations between SP140 expression and immune cells’ infiltration across 26 TCGA solid tumor types, using the TIP database. Pink and green denote positive and negative correlations, respectively. The number of TCGA cohorts correlated positively (pink) or negatively (green) is presented on the right panel. (**F**) The number of TCGA cohorts showing positive (pink) or negative (green) correlations between the expression of selected markers and immune cell infiltration, using the TIP database.

**Figure 3 ijms-24-00549-f003:**
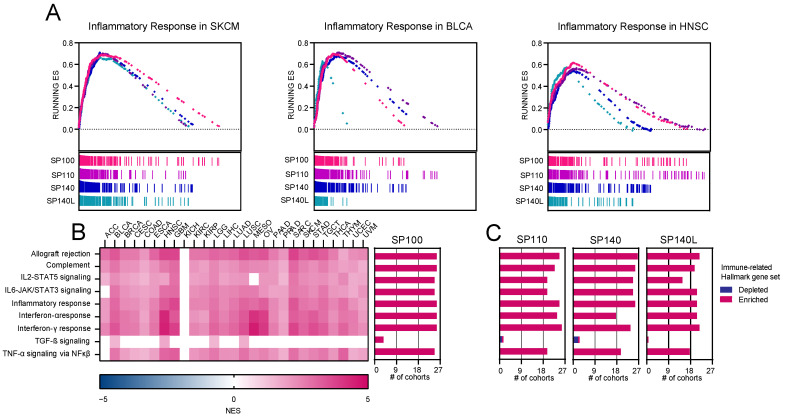
Stemness signature enrichment in the transcriptome profiles associated with the expression of SP family members (*SP100*, *SP110*, *SP140*, and *SP140L*). For AIRE and ATAD2-associated transcriptome profiles, the gene-set-size threshold was not passed during the GSEA. The GSEA was performed by using significantly differentially expressed genes (*p* < 0.05, FDR < 0.05) in TCGA patients divided into low-expressing or high-expressing BrD cohorts (using the mean expression of each gene as a cutoff). (**A**) The enrichment plots of Inflammatory Response Hallmark term in transcriptome profiles associated with *SP100*, *SP110*, *SP140*, and *SP140L* in SKCM, BLCA and HNSC. Gene ranks for *SP100* (magenta), *SP110* (purple), *SP140* (blue), and *SP140L* (green) associated transcriptome profiles are presented at the bottom of the enrichment plots. (**B**) The heatmap of NES of immune-related Hallmark gene sets among 26 solid tumor types for SP100 with a summarized number of cohorts. Color of the heatmap denotes either increase (magenta) or decrease (dark blue) of NES. # of cohorts – the number of TCGA cohorts with either positive (magenta) or negative (dark blue) association with immune-related Hallmark terms in high-expressing BrD groups. (**C**) Summarized number (#) of cohorts with either increased (magenta) or decreased (dark blue) NES of immune-related Hallmark gene sets for *SP110*, *SP140*, and *SP140L*.

**Figure 4 ijms-24-00549-f004:**
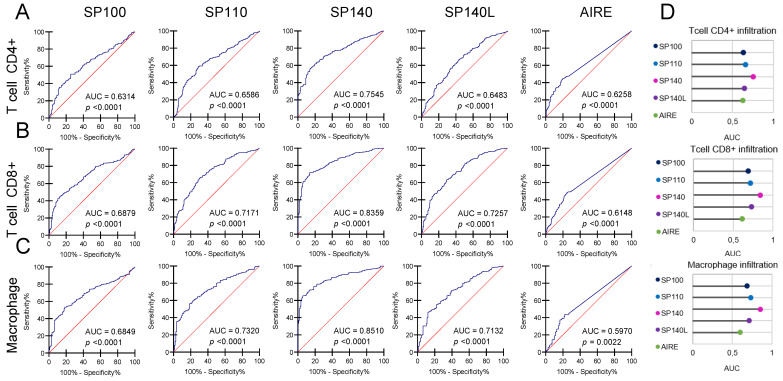
Survival prognosis based on immune-cell infiltration for SP family (*SP100*, *SP110*, *SP140*, and *SP140L*) and control (*AIRE*) gene in SKCM. The ROC curves show the prognostic significance of SP family members and control genes related to the infiltration of relevant immune cell populations: T cell CD4+ (**A**), T cell CD8+ (**B**), and macrophage (**C**). The area under the curve (AUC) was calculated for the ROC curve. (**D**) The AUCs for *SP140* exhibit the highest values for all tested immune cell populations in SKCM. The results for *ATAD2* control genes are not shown due to lack of statistical significance (*p* > 0.05).

**Figure 5 ijms-24-00549-f005:**
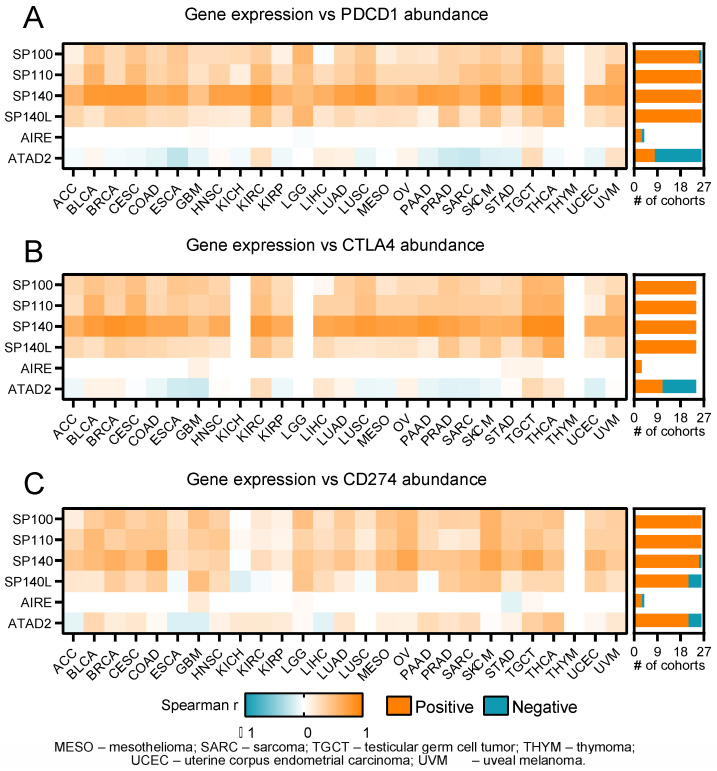
The heatmap of Spearman’s correlation between SP family (*SP100*, *SP110*, *SP140*, and *SP140L*) and control genes’ (*AIRE* and *ATAD2*) expression and immunoinhibitors’ ((**A**) PDCD1, (**B**) CTLA4, and (**C**) CD274) abundance among 27 tumor types with the number (# of cohorts) of positively (orange) or negatively (aquamarine) correlated cohorts. Color of the heatmap denotes either increase (orange) or decrease (aquamarine) rho value with statistical significance (*p* < 0.05).

**Table 1 ijms-24-00549-t001:** Classification and function of BrD-containing proteins.

Group Number	Function	Proteins
I	Histone acetyltransferase	PCAF, GCN5L2, p300/CBP, TAF1, TAF1L, BRPF1A/B, BRPF2 (BRD1), BRPF3, BRD8 (SMAP)
II	Histone methyltransferase	ASH1L, MLL
III	Chromatin remodeling factor	SMARCA2A/2B, SMARCA4, PBRM1, BRD7, BRD9, BAZ1A, BAZ1B, BAZ2A, BAZ2B, BPTF, CECR2
IV	AAA ATPase protein	ATAD2, ATAD2B
V	BET family transcriptional coactivator	BRD2, BRD3, BRD4, BRDT
VI	E3 SUMO/ubiquitin ligase	TRIM24, TRIM28, TRIM33A/B, TRIM66
VII	SP-family proteins of PML nuclear bodies	SP100, SP110, SP140, SP140L
VIII	Transcriptional corepressor	ZMYND8, ZMYND11
IX	WD-repeat protein	BRWD1, BRWD3, PHIP

## Data Availability

Only publicly available data were used in this study, and data sources and handling of these data are described in the Materials and Methods and in the Appendix A. Further information is available from the corresponding author upon request.

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
