# Peer review of "It Runs in the Bromodomain Family: Speckled Proteins (SP) Play a Role in the Antitumor Immune Response in Solid Tumors"

_ijms, 2022, doi:10.3390/ijms24010549_

Round 1
Reviewer 1 Report
The authors have produced a comprehensive manuscript that provides new information about the role of bromodomain proteins and immune cell infiltration in solid tumors. The lack of any in vivo experiments is an obvious omission to test these findings. However, overall the study offers compelling data that can be used to understand the nature of these interaction in the tumor microenvironment.
Author Response
We appreciate the time and effort that the Reviewer has dedicated to revising our manuscript. We are grateful for the comment.
Reviewer 2 Report
The article presented by Rosochowicz et al. presents a series of bioinformatics analyses based on transcriptomic data showing a correlation between SP family proteins and some determinant factors in TME such as immune cell infiltration. The study is well-designed and the results and conclusions are easy to understand and follow by the reader.
However, after revising the manuscript, this reviewer has some doubts about some of the authors' approaches and has detected some errors that should be corrected by the authors before the publication of the article in press.
As for the introduction, the content is correct, since it explains to the reader the role of the proteins on which the authors focus their study. In the case of the end of the introduction, where the authors state the hypothesis, they also end up reporting the conclusions of their study. The authors should eliminate lines 135-138 from the introduction because, from a formal point of view, they are incorrect in the introduction section.
In the first results section "2.1", as explained in materials and methods, eligibility criteria have been applied for the analysis of these cell types. The authors should consider, in addition to the correlations shown, whether there is any correlation by tumor type studied since no comment is made in this section. Also indicate to the authors that the figure caption is very large and is cut between pages, therefore, it is difficult to know what type of tumor is reflected in the graph. I suggest to the authors that they include a legend within the figure itself, which will make the figure caption smaller and will also improve the quality of the follow-up of the figure.
Regarding section 2.2, only the analysis of the SKCM tumor is included. On the one hand, in line 184 it is necessary to include a reference that supports the authors' statement that this type of tumor was chosen. In addition, this reviewer believes it is necessary to include the analysis of a tumor type other than SKCM in which the trend in TME is the opposite of SKCM. This would reinforce the results presented by the authors. Also, note that the figure is indicated in the figure caption as "Figure 1" and is "Figure 2". It is well-cited in the text.
In the case of section 2.3, the authors now analyze three cell types but provide no justification for their choice. The inclusion of a justification of the choice is needed. In this case, in Figure 3, it is not well indicated to which panel (A, B, C or D) each description corresponds.
The authors should improve the quality of Figure 4 as it is difficult to read the AUC values.
Finally, in section 2.5, where 27 cell types are analyzed, the types are not detailed anywhere in the manuscript.
In relation to the discussion, although it is very extensive and detailed, the authors should include aspects of the limitations of their study for completeness.
Lastly, regarding the statistical analysis, this reviewer considers that the description of materials and methods is very scarce. The authors should include more details on the statistical methods used, such as Spearman or ROC curves.
Round 2
Reviewer 2 Report
The authors have successfully addressed the requirements.